# Molecular epidemiology of *Ascaris lumbricoides* following multiple rounds of community-wide treatment

Toby Landeryou [1] ✉, Rosie Maddren[2], Jack Hearn [1,4], Mahlet Belachew [3], Santiago Rayment Gomez[2], Ewnetu Firdawek Liyew[3], Kathryn Forbes[2], Birhan Mengistu[2], Scott P. Lawton[1,4], Jude Eze[1], Geremew Tasew[3], Ufaysa Angulo[3] & Roy Anderson[2,4] ✉

Control and elimination of the parasite *Ascaris lumbricoides* relies on mass drug administration (MDA) using a limited number of anti-helminthics. Whilst these programs have reduced the infection intensity and prevalence within many endemic regions, patterns of transmission remain poorly understood. Reinfection commonly occurs following cessation of treatment due to the absence of acquired immunity post infection. Here, we utilise genomic data to understand parasite transmission within and between households in a community and the genomic impact of repeated MDA. We sequenced 54 whole-genomes from Ascaris worms obtained from individuals in a longitudinal cohort epidemiological study of transmission and drug treatment extending over 6 years. We found that fine-scale population structure exists in spatially distinct clusters of infected individuals with reinfection occurring within or between geographically close households. This observation helps inform the policy for future control in low prevalence settings suggesting more targeted treatment of infection hotspots. We found evidence of positive selection acting on members of gene families previously implicated in reduced drug efficacy but detected no impactful variants. As efforts to eliminate *A. lumbricoides* intensify, our study provides a foundation for genomic surveillance to help identify both who infects whom and the impact of repeated drug treatment.

The soil-transmitted helminths (STHs) are a group of intestinal parasites designated by the World Health Organisation (WHO) as the aetiological agents of a neglected tropical disease. There are an estimated 1.5 billion individuals infected with at least one intestinal nematode infection globally, cumulatively resulting in 750,000 DALY's a year, with disability peaking in children 5–9 years old[1,2]. *Ascaris lumbricoides* is one of the most prevalent

species of STH, with infection occurring through the ingestion of embryonated eggs from contaminated soil, food and/or water. Ascariasis is estimated to affect ~819 million people globally[3], predominantly school-age children (SAC) who, on average, harbour a greater number of parasites than adults. Treatment is delivered via mass drug administration (MDA) campaigns using the anthelminthic drugs albendazole and mebendazole within

[1]Centre for Epidemiology and Planetary Health, School of Veterinary Medicine, Scotland's Rural College, Inverness IV23JX Scotland, UK. [2]Department of Infectious Disease Epidemiology, School of Public Health Building, Imperial College London, London, UK. [3]Malaria, Neglected Tropical Disease Research Team, Bacterial, Parasitic, Zoonotic Diseases Research Directorate, Ethiopian Public Health Institute, Addis Ababa, Ethiopia. [4]These authors jointly supervised this work: Jack Hearn, Scott P. Lawton, Roy Anderson. ✉e-mail: toby.landeryou@sruc.ac.uk; roy.anderson@imperial.ac.uk

communities on an annual or biannual basis, dependent on the intensity of transmission[4].

Over the past three decades, these chemotherapeutic control programmes have been effective at reducing the prevalence and intensities of infection to low levels in many endemic regions (http://www.espen.afro.who.int/diseases/soil-transmitted-helminthiasis). Following these successes, the WHO set a goal to achieve elimination as a public health problem of *A. lumbricoides* (defined as <2% heavy or moderate intensity infections) in pre-SAC and SAC in all 78 countries with endemic infection and eliminating transmission in selected regions by 2030[5]. The currently recommended treatment strategy only targets SAC and pregnant women; however, to reach these goals, some recently implemented deworming projects have expanded treatment strategies from targeting only school-based MDA (sMDA) to community-wide MDA (cMDA)[6–8]. To achieve transmission break at a community and regional level, prevalence must be driven down to 2%, this is supported through numerous mathematical models of STH transmission and control, and several large-scale epidemiological studies[7,9–11]. Untreated infected individuals (untargeted adult population and untreated SAC), with high worm burdens, act as reservoirs of infection for recently treated and STH-cleared individuals, and hence, community-wide treatment is required to eliminate all transmission within defined populations[12]. Furthermore, prior infection does not confer strong acquired immunity. Therefore, especially in areas with poor access to adequate water, sanitation and hygiene (WaSH) infrastructure, repeated reinfection is common, which frequently results in rapid bounce-back of local infection after cessation of MDA[10,13–15]. Where treatment coverage is not delivered at a consistently effective level to the target populations, a constant cycle of treatment and reinfection will remain.

The distribution of *A. lumbricoides* infection in endemic populations is over-dispersed or aggregated, which is well described by the negative binomial probability distribution, where most harbour none or a few worms and a few are infected by many[16]. This aggregation is measured by $k$, the negative binomial parameter. As chemotherapeutic pressure increases over repeated rounds of MDA, the degree of worm aggregation also tends to increase (very low $k$ values[17,18]), creating hotspots of infection in the community, which are challenging to identify through routine surveys and therefore treat. Those heavily infected tend to be predisposed to infection.

Accurately measuring changes in parasite population size and structure within these low-prevalence communities is crucial to an understanding of the current and future impact of expanded MDA. Tracking how these few individuals with the remaining worms acquire infection and sustain infection in the wider community is key to the successful 'end game' STH control in regions that have had decades of MDA. The remaining pockets of infection may be attributed to many factors: systemic non-compliance of households to treatment, poor WaSH status, or social and cultural factors influencing exposure to infection[19]. In understanding the epidemiology of infectious diseases, establishing 'who infects whom' can be defined as the study of transmission dynamics of a pathogen and the patterns and processes by which infectious diseases spread within a population[20]. Within the context of *A. lumbricoides*, understanding of 'who infects whom' involves identifying the sources of infections and understanding how the parasite is transmitted from one individual to another. From a treatment policy perspective, once prevalence is driven down, a key question is which individuals or households play a central role in sustaining transmission or are considered superspreaders.

Establishing 'who infects whom' via molecular epidemiological studies based on worm expulsion and whole-genome sequencing (WGS) is an obvious step in seeking to understand how transmission is sustained. WGS of expelled worms serves to reveal genetic diversity, and associated parasite population structure and relatedness. This information can be used to infer recent demographic changes[21], who infects whom by relatedness analyses[22], the efficacy of interventions on diversity, and the associated impact of repeated treatment on single drug use[23–25]. While a number of recent studies have looked at the population genetics of *A. lumbricoides* in endemic regions[26–29], the majority of these have employed a limited number of molecular markers with a focus on the relatedness of *A lumbricoides* to parasites in domestic animals such as pigs (e.g. *Ascaris suum*)[27].

In the context of infection control, the presence of multiple parasite genetic clusters within the host community and the relatedness between them may help identify transmission foci. Multiple sites of transmission, such as a local school serving many villages, or indeed specific households, can in principle be identified by molecular epidemiological studies based on WGS. While this approach has been adopted at a global geographic scale[30,31], to date, it has not been employed within a single community linked to longitudinal epidemiological studies of infection and control.

To elucidate the genomic impact of long-term MDA pressure on endemic communities and to gain insights into the genomic diversity and gene flow amongst A. *lumbricoides* populations in an endemic community, we performed WGS on parasites collected during a worm expulsion survey in November 2022. We collected these worms as part of the Geshiyaro project[11], which is designed to measure the success of MDA and WaSH infrastructure improvements delivered with behaviour change communication sensitisation (Fig. 1). The Geshiyaro

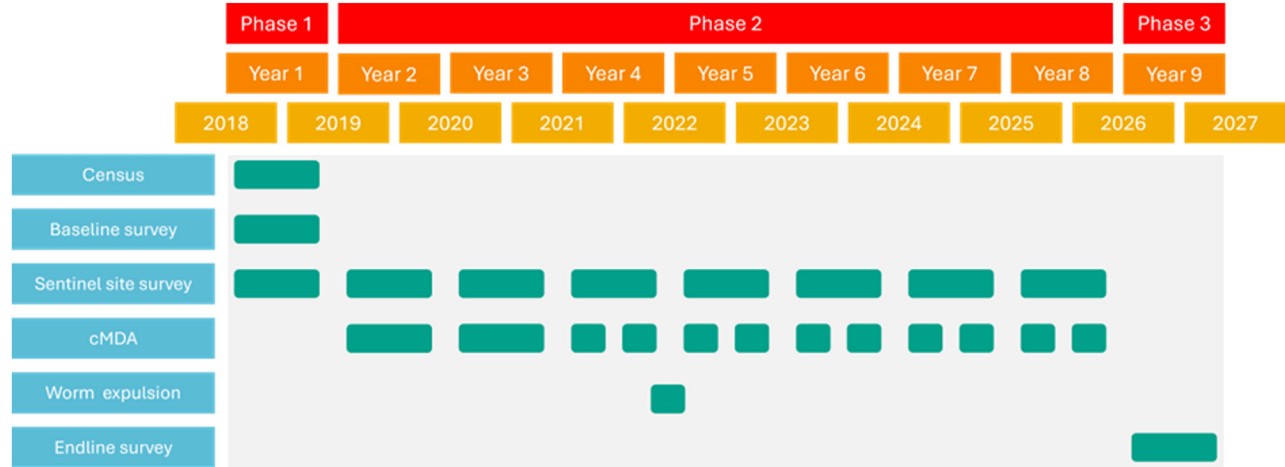

**Fig. 1 | Longitudinal Gantt chart of M&E activities.** Plot describes the monitoring and evaluation activities that have taken place across the entire Geshiyaro project in the Korke Doge community, including the timing of worm expulsion activity taking place after four rounds of MDA.

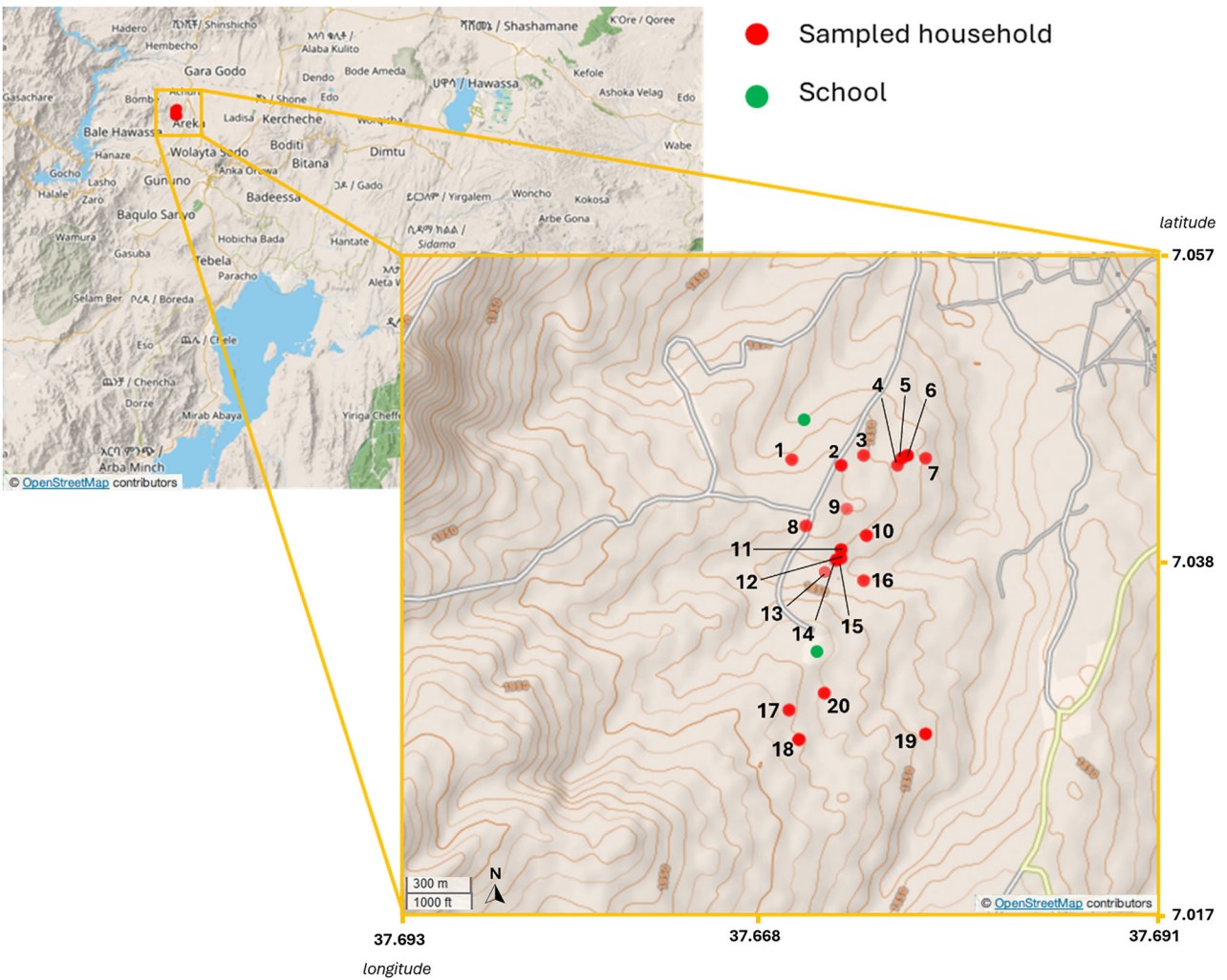

**Fig. 2 | Map of sampled households.** Map describes the sampled households as part of this study in Korke Doge kebele in the Wolaita zone of Southern Ethiopia. Households sampled are denoted in red while school locations are highlighted in green. Household numbers are included alongside location points. Map tile © OpenStreetMap contributors, licensed under the Open Database License (ODbL).

project's key objective is to interrupt STH transmission in the Wolayita region of Ethiopia. Using individual-level epidemiological data alongside genomic data from chemotherapeutically expelled worms, we report the genetic structure of *A. lumbricoides* and patterns of transmission within and between households and villages. We also determined whether there are observable changes in STH diversity that could be attributed to repeated albendazole treatment. We also assessed the presence of variants which have been implicated in reduced albendazole efficacy[32–34].

To establish the effectiveness of the treatment programme, our analysis was disaggregated by age and drug compliance groupings (the fraction of the population who repeatedly take drug treatment) defined in the Geshiyaro control programme[11,35]. We utilised household GPS data of participants to understand the presence of spatially explicit patterns of genetic associations between worms collected across the community (Fig. 2)[36]. This study represents the first combination of a longitudinal epidemiological study alongside individual-level genomic data of *A. lumbricoides* to ascertain who infects whom and the genomic impact of repeated drug treatment.

## Results

### Changes in the prevalence and intensity of infection over time

Prevalence across the Korke doge cohort was established through the identification of *A. lumbricoides* faecal egg-counts via microscopy,

following a 2-day, duplicate slide Kato-Katz methodology described in a published protocol paper[11]. The baseline prevalence of *A. lumbricoides* in Korke Doge was 38.6% (95% CI: 30.7–46.6%) in Year 1 (2018), which decreased to 9.27% (95% CI: 4.63–13.9%) in Year 5 as illustrated in Fig. 3a. Prior to the worm collection, the kebele received four rounds of cMDA at a compliance rate (treatment swallowed) of 89.7%, 83.1%, 84.4% over three annual rounds, and a compliance rate of 87.7%, 73.9%, 60.0%, 68.9% over four subsequent biannual rounds.

The proportion of participants who remained infected between survey years decreased from 16.4% to 4.1%, as demonstrated in Table 1. The level of predisposition to no, light or heavy infection, as measured by Kendall's tau ranking correlation statistic, in all the sentinel site cohorts in the Geshiyaro study increased from 0.18 (Year 1 to Year 2, $p < 0.001$) to 0.27 (Year 4 to Year 5, $p < 0.001$). For individuals in Korke Doge, predisposition also increased from a Kendall's tau value of 0.19 (Year 1 to Year 2, $p < 0.05$) to 0.23 (Year 4 to Year 5, $p < 0.01$). Higher levels of predisposition to heavy infection were seen in individuals who were partially treated, compared with those who were always treated (Table 2).

### Worm burden

A total of 102 individuals were sampled across the community over 5 days following community-wide albendazole treatment. Of these 102 individuals, 54 individuals were found to expel at least one adult worm,

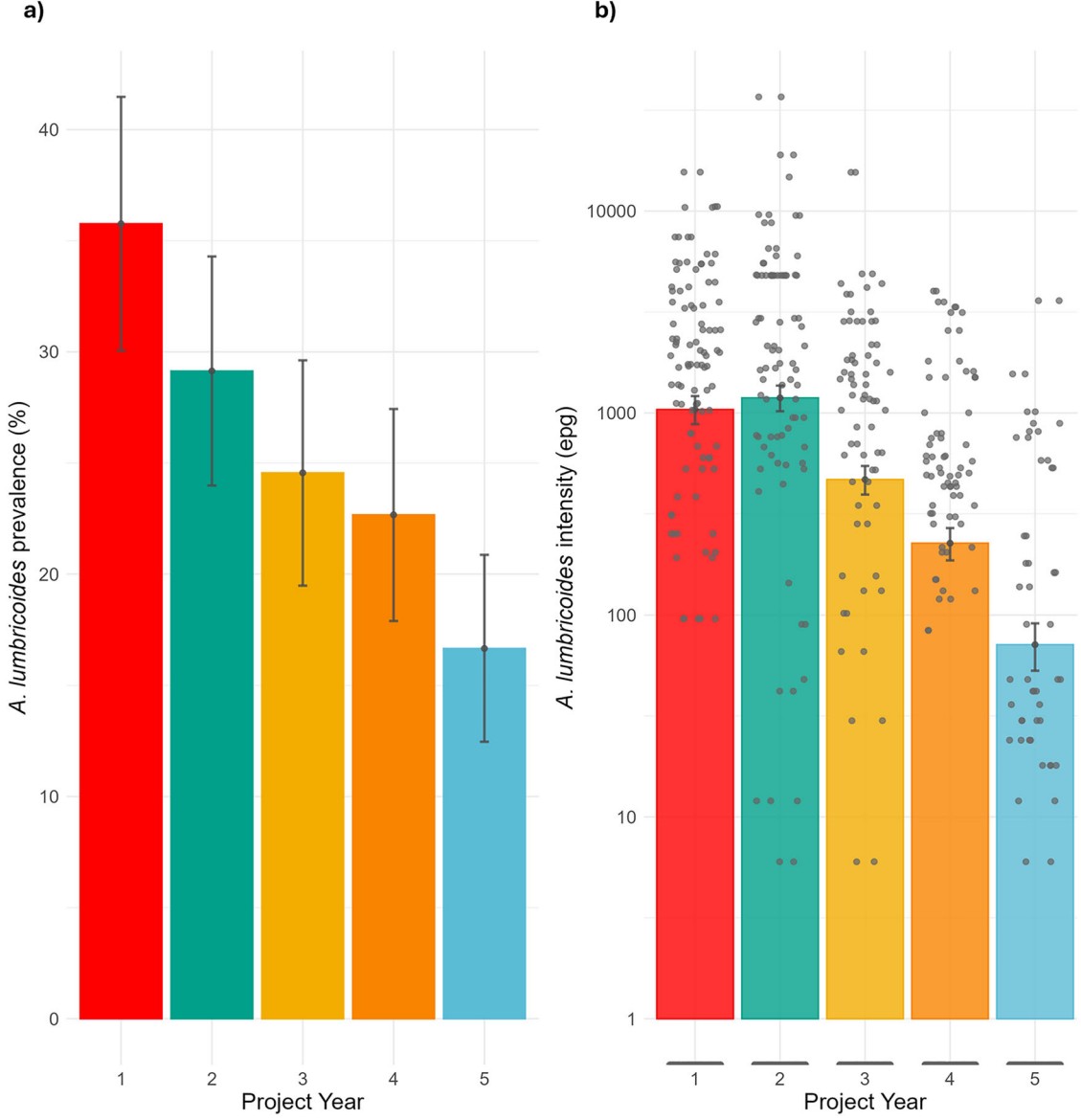

**Fig. 3 | Description of longitudinal epidemiological data in cohort.** Change in **a** prevalence and **b** intensity (right-hand axis) of *A. lumbricoides* in Korke Doge over 5 years of project implementation from 2018 to 2022. Error bars show the 95% confidence limits. All infections are classified as low intensity according to WHO guidelines (1–4999 eggs per gram). Participant sample size for each year: Year 1 ($n = 274$), Year 2 ($n = 302$), Year 3 ($n = 281$), Year 4 ($n = 300$), Year 5 ($n = 306$). Prevalence errors are presented as mean values ± SD, with a central point showing the mean value. Intensity negative binomial error bars are presented as the mean ± SD, with a central point showing the mean value.

with 222 worms expelled in total (Supplementary Data 1). Worm count per person showed a slight increase in Pre-SAC and SAC individuals, but this was not statistically significant (Supplementary Fig. 1). Of the 54 individuals that expelled worms, 31 were from non-compliant individuals, 11 from semi-compliant and 8 from fully compliant (Supplementary Data 1).

### Whole-genome sequencing of individual worms

Alignment of sequence reads to the reference *A. lumbricoides* genome revealed high-sample mapping rates (median 87.99%) of reads mapped with the mapping quality MQ > 40 (Accession numbers included in Supplementary Data 2). Reads mapped to the majority of the *A. suum* reference genome[37] (median of 94.23% of bases covered across all samples), although the median read depth was highly variable across the total sample of worms (7.33-49.34X). No relationship was observed between specific regions or chromosomes which displayed consistently low coverage. Across all worms, a single worm was taken per

individual for a final analysis dataset of 54. The final variant dataset comprises 3,692,001 single-nucleotide polymorphisms (SNPs) and 280,447 indels (including mixed SNP/indel variant sites) from 54 adult worms.

### Local parasite population genetic structure

The population structure across the 54 worm sample set is depicted by a variety of analyses in Fig. 4. We generated a maximum likelihood phylogeny (Fig. 4a) and a principal component analysis (PCA) using a subset of 1,000,000 unlinked autosomal variants (Fig. 4b). A cross-validated PCA indicated that three genetic clusters exist across the infected households in the sampled community (Fig. 4b, c), this was further confirmed through likelihood values demonstrated for number of subpopulations ($K$) (Supplementary Fig. 3). Specific differences in inbreeding coefficients are observed across households 1, 3, 12, 13, 14, 15, 17, 19 and 20 in Fig. 5 indicating reduced gene flow between these households and others within the community. Pairwise measures of

**Table 1 | The year-on-year change in infection status in all surveyed individuals with the Korke Doge kebele**

| Compared years | Total | Gained infection | Kept infection | Lost infection | Never infected |
|---|---|---|---|---|---|
| y1y2 | 111 | 18 (16.4) | 17 (15.3) | 21 (18.9) | 55 (49.6) |
| y2y3 | 112 | 13 (11.6) | 12 (10.7) | 18 (16.1) | 69 (61.6) |
| y3y4 | 99 | 14 (14.1) | 5 (5.05) | 14 (14.1) | 66 (66.7) |
| y4y5 | 111 | 10 (9.01) | 8 (7.21) | 18 (16.2) | 75 (67.6) |

Numbers of 'Never Infected', 'Kept Infection', Gained Infection and 'Lost Infection' are disaggregated by counts and (percentage) of individuals under the infection designation.

**Table 2 | Predisposition to *A. lumbricoides* infection as measured by the correlation coefficient, Kendall's tau: a rank correlation, non-parametric procedure**

| | | y1y2 | y2y3 | y3y4 | y4y5 | y1y5 |
|---|---|---|---|---|---|---|
| All Geshiyaro Sites | Overall | 0.180 P 0.00 | 0.253 P 0.00 | 0.215 P 0.00 | 0.253 P 0.00 | 0.083 P 0.1 |
| | Always treated | 0.065 P 0.13 | 0.134 P 0.2 | 0.170 P 0.31 | – | – |
| | Partially treated | 0.21 P 0.004 | 0.259 P 0.006 | 0.178 P 0.00 | 0.111 P 0.0025 | 0.098 P. 0.03 |
| Korke Doge kebele | Overall | 0.194 P 0.043 | 0.277 P 0.00 | 0.117 P 0.073 | 0.233 P 0.005 | 0.213 P 0.03 |
| | Always treated | 0.098 P 0.26 | 0.128 P 0.07 | 0.118 P 0.08 | – | – |
| | Partially treated | 0.177 P 0.058 | 0.269 P 0.005 | 0.105 P 0.07 | 0.259 P 0.03 | 0.206 P 0.06 |

Correlation is measured on a scale from –1 to 1, where values close to 1 indicate a high level of predisposition to infection, whereby those who were infected were more likely to be reinfected again. *P* values of significance are given as *P* < 0.001, *p* < 0.01, *p* < 0.05, or *p* > 0.05.

fixation revealed low levels of diversity across all households (Supplementary Data 3). Across the community, we established summary values of HO = 0.212, HE = 0.405, $p = 0.395$, and FIS = −0.121 for the full dataset (Supplementary Fig. 4). We characterised the population structure at different spatial scales using a subset of 1,000,000 autosomal variants. These were filtered to remove variants in strong linkage disequilibrium. PCA revealed that 34% of the total variance was explained through the first two components. Phylogenetic analysis revealed no real discernible structure within parasites across Korke Doge apart from the three distinct genetic groupings of the parasite shown in Fig. 5a, b. Genome-wide estimates of median nucleotide diversity within parasite populations revealed low levels of differentiation ($\pi = 0.02$, $d_{XY} = 0.09$; Supplementary Fig. 4). Parasite populations within households showed approximately equivalent levels of nucleotide diversity ($\pi = 0.01$, $d_{XY} = 0.05$) (Supplementary Fig. 4).

### *Ascaris lumbricoides* population size within the community following multiple rounds of community-wide MDA

We estimated an effective parasite population size of 201.3 (95% CI 197.1–205.5) across the entire community. The dataset was further disaggregated by an individual's compliance with the annual treatment programmes, where compliance is taken to mean the swallowing of the offered treatment. The effective population of worms within different compliance groups was as follows: fully compliant 29.9 (95% CI 21.3–38.2), semi-compliant 98.5 (95% CI 80.7–106.3) and non-compliant 121.3 (95% CI 115.8–136.8). Additionally, the community was stratified into defined age groups; namely, Pre-SAC (2–4 years), SAC (5–14 years), adolescents (15–20 years), young adults (20–35) and adults (36+). The effective population size of parasites in these age groups were estimated to be; Pre-SAC 88.2 (95% CI 54.8–121.6), SAC 135.6 (95% CI 119.5–151.7), Adolescents 45.2 (95% CI 36.1–51.3) and Adult 57.5 (95% CI 40.9–64.1) (Supplementary Fig. 4). In summary,

most worms are predicted to be in SAC, and within that group, the majority are in those non-compliant to treatment (Supplementary Data 1).

Genome-wide allele frequency patterns allow the determination of recent reductions in population size and have been demonstrated as a reliable metric for establishing demographic size change in helminth populations[25]. Analysis of one-dimensional site frequency spectra across the Korke Doge sample does not indicate an abundance of rare (singleton and doubleton) alleles which suggests the parasite population effective population size is small and declining (Supplementary Fig. 5a). Median genome-wide Tajima's D estimates were positive across all compliance groups which suggests a recent population constriction most likely influenced by the past four rounds of MDA (Supplementary Fig. 5b). Demographic size change indicated that the effective population size of parasite populations over time, revealing current populations are a fraction of the size of the historical populations from which they were derived (Supplementary Fig. 5c).

### Community-wide spatial population genomics

We identified signals of population differentiation using methods that aim to identify fine-scale genetic structuring. Co-ancestry heatmaps between individual worms (Fig. 5) highlighted reduced gene flow between worms from within households compared to between. Pairwise comparisons among these households reveal a pattern of co-ancestry with increased gene flow between closely located households and decreased gene flow with worms in the more distant households.

Plotting values from the first axis of the sPCA analysis indicated a break in connectivity across these spatially distinct clusters. In Fig. 6a, grey-scale squares indicate households with spatially discriminated clusters of worms. Figure 6b shows the results of spatial PCA analysis, which are a composite measure of both genetic diversity (variance) and spatial structure (autocorrelation through Moran's I). $\lambda_1$ scores from spatial PCA show a distinct spatial structure across households within sampled community (Fig. 6). Colour assigned spatially distinct clusters indicate that neighbouring households demonstrate reduced variance between them, with increasing variance occurring within households at greater distance. Results have been plotted across the Delauney triangulation connection network used to calculate spatial weightings, with each node within the network representing a sampled household. Plotting of eigenvalues resulting from sPCA revealed a pattern of local-scale structuring (Supplementary Fig. 6). Plotting of interpolating lagged principal scores show a map of genetic clines between the sampled households in the community (Supplementary Fig. 7). Spatial clustering was confirmed through spatial autocorrelation output from Moran's I (Supplementary Fig. 8).

### Evidence of positive selection in a region that has experienced long-term MDA

A number of genome-wide methods were used to identify regions of the genome undergoing positive selection in worms from Korke Doge following multiple rounds of MDA. Multiple regions were identified by the integrated haplotype score (iHS) test as being under strong positive selection in Korke Doge (Fig. 7a, b). Overall, we identified 69 non-redundant regions with extreme iHS scores (Supplementary Data 3). From these 69 regions, we found 14 regions that indicated both elevated iHS and $F_{ST}$ values. We also determined whether any of these selected regions intersected with regions of reduced genetic diversity (Supplementary Fig. 9). These regions were defined as encompassing genes which contained at least one variant in one or more worm samples[38]. Candidate regions of extreme iHS scores spanned 91 protein-coding loci with known functionality, with a further 38 classified as hypothetical proteins (Supplementary Data 3).

One region that displayed strong signals of selection was located within chromosome 1. Across chromosome 1 there was three candidate regions (CR1 1.13–1.15 Mb, CR2 2.58–2.6 Mb, CR3 7.27–729 Mb and CR4

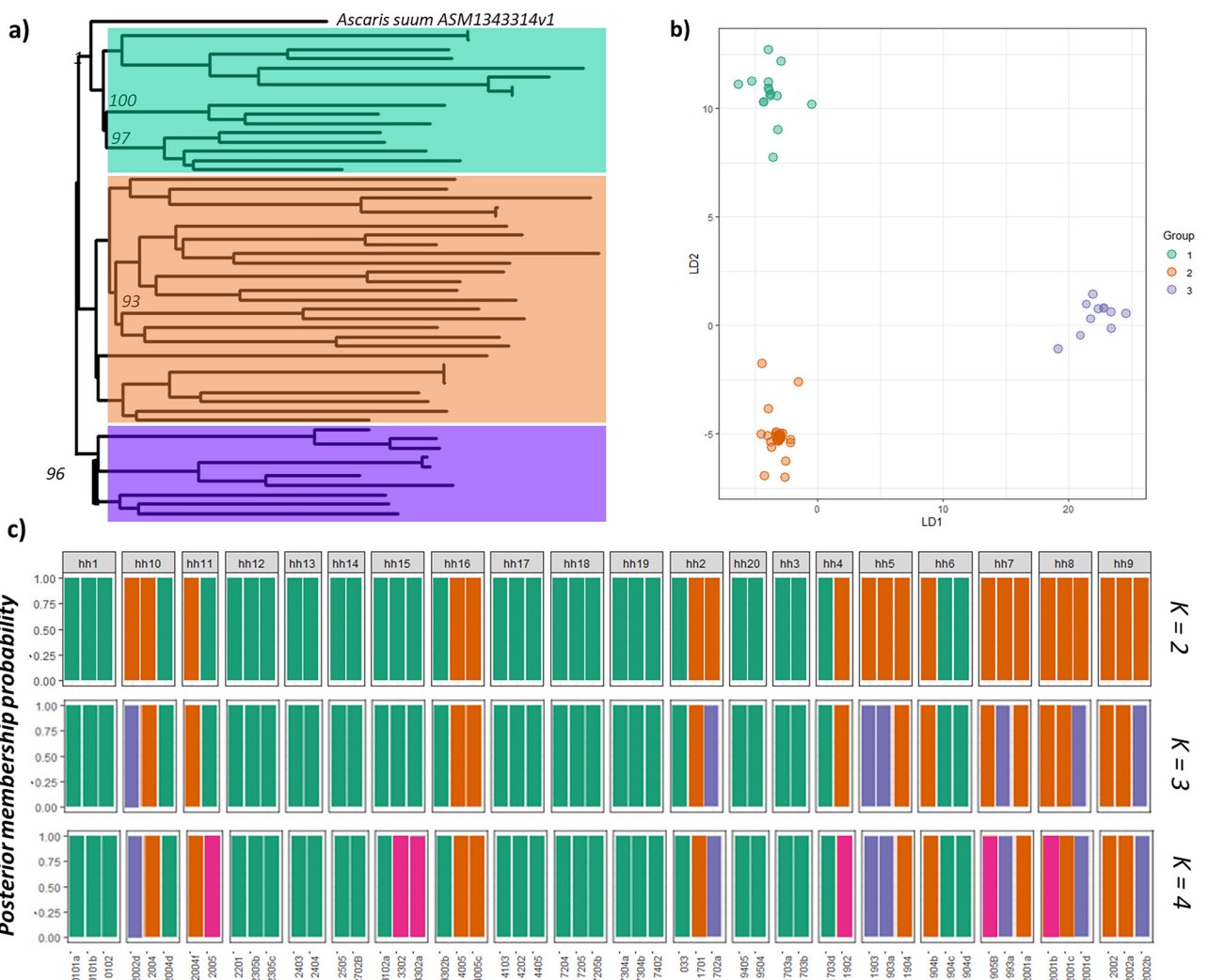

**Fig. 4 | Principal component analysis (PCA) of genetic differentiation across the 54 *A. lumbricoides* sampled for WGS. a** Neighbour-joining phylogeny showing relatedness between samples, topology groups are coloured according to principal component clustering. **b** Principal components 1 and 2 accounting for ~34% of total variance. c cluster assignment illustrating the cluster assignment per individual per household. **c** ADMIXTURE plots illustrating the population structure, assuming 2–4 populations are present (*K*), using 10-fold cross-validation and standard error estimation with 500 bootstraps. *Y*-axis values show admixture proportions for different values of *K* (*K* = 2–4), and each colour indicates a different population.

17.52–17.55 Mb) which demonstrated increased levels of localised selection (Fig. 7). Across these regions CR3 and CR4 demonstrated significantly elevated iHS values, increased $F_{ST}$ and decreased relative nucleotide diversity. Within the region on chromosome 1, we identified a single locus containing the beta-tubulin gene (*AgB04_g300*), which shows three non-synonymous variants at low to mid frequencies (0.11–0.42, Supplementary Data 3) at amino acid positions 7, 9 and 67. Additional beta-tubulin genes were within localised regions of selection in Chr X3 (7.87–7.877 Mb). However, no non-synonymous mutations were identified in this region (Supplementary Data 4). The beta-tubulin families are implicated in reduced susceptibility of *A. lumbricoides* to anti-helminthics within the benzimidazole family[32,39,40]. There was no evidence for specific mutations at three codon positions[41,42] previously linked to resistance against albendazole in *A. lumbricoides*[43,44].

Further evidence of positive selection was found across multiple regions of chromosome 10 (CR20 1.14–1.16 Mb, CR21 3.65–3.67 Mb, CR22 3.89–3.91 Mb, CR23 4.81–4.83 Mb, CR24 4.95–4.99 Mb, CR25 6.22–6.24 Mb) which encompassed 19 genes (7.47–7.72 Mb). A variety of other genes in candidate regions encode non-synonymous mutations, including a microtubule-associated protein (*AgR001_g156*),

dynactin, annexin and transcription factors, among others (Supplementary Data 3). A potassium voltage-gated ion channel protein was located within candidate region 27 (7.59–7.63 Mb).

## Discussion

To sustainably interrupt transmission of STH, there needs to be a better understanding of how these parasites maintain circulation in low-prevalence communities. We performed WGS of individual *A. lumbricoides* parasites collected within a longitudinal epidemiological study of infection and treatment, within a single community. This is the first time that such a molecular epidemiological analysis has taken place within an expanded, community-wide treatment programme for the control of STH. The study of one village and a relatively small sample of worms where WGS was possible (54 worms from an estimated total population of over 200 worms (~25%)) revealed several insights into transmission. The epidemiological data collected here have expanded upon the findings regarding the prevalence and intensity of infection across the Korke Doge community as part of the mid-point survey within the control programme[35]. The continued collection of infection data within the community indicates the persistent low prevalence of infection taking place in Korke Doge,

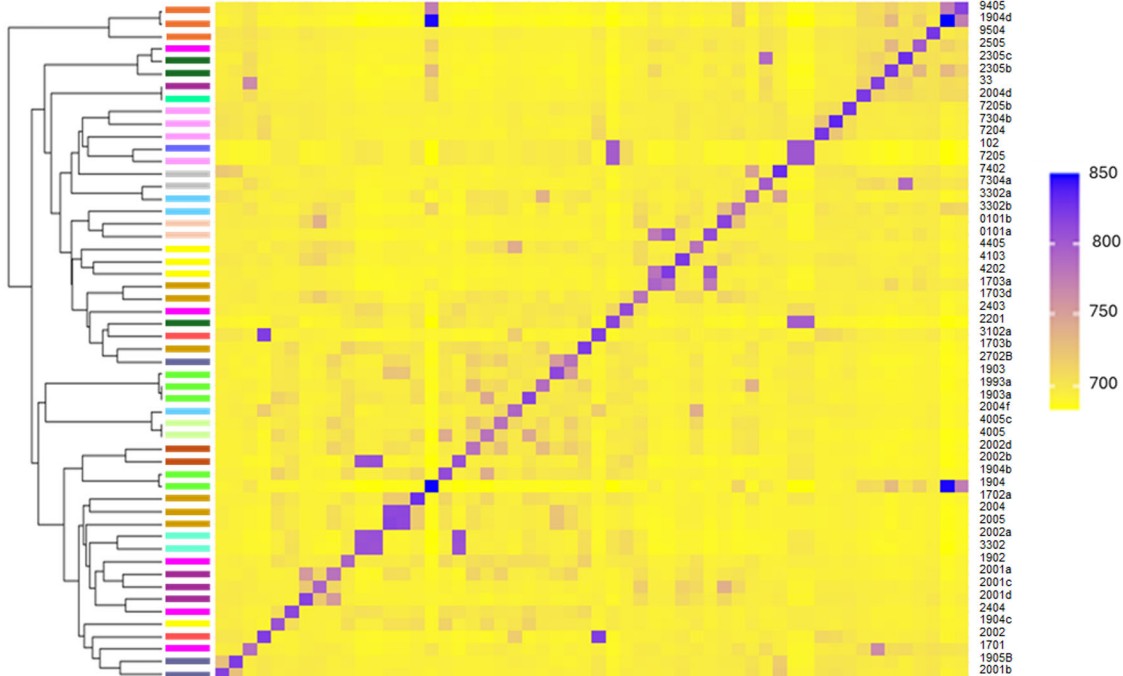

**Fig. 5 | Co-ancestry heatmap across individuals within community.** Co-ancestry heatmap was generated through fineSTRUCTURE analysis. Samples grouped along the heatmaps' diagonal have common shared co-ancestry histories and pairwise comparisons outside the diagonal indicate level of co-ancestry between individuals. Lighter yellow represents lower shared co-ancestry and shades of purple indicate progressively higher shared co-ancestry. Colours of corresponding bands alongside the dendrogram indicate household membership in which each worm was sampled.

mirroring the global status of STH. First, genome data suggest that *A. lumbricoides* displays fine-scale population structure within a village with endemic infection (Figs. 4 and 6). Second, multiple infective foci are present within the study community, and third, individuals within the village are not being infected by a relatively homogenous randomly mixed population of worms.

The data suggests that transmission primarily occurs at the household level rather than a single source community-wide infection. Genetic relatedness is strongest between closely located households rather than those more widely spatially distributed. The focal infective nature of *A. lumbricoides* infection within endemic regions and communities has been suggested from parasitological studies of infection patterns in Nepal[30], Venezuela[45] and Guatemala[46]. PCA across our spatial network indicates a strong signal for local-scale structuring, inferring that genetic clusters are spatially explicit. The increase in homozygosity within these spatially explicit groups of worms suggests that directional selection will exhibit greater efficiency in driving an increase in potentially advantageous alleles across populations, such as drug-resistance genes within a population of frequently treated worms[47]. As prevalence is driven down by numerous rounds of treatment, understanding these patterns of spatial clustering would permit a more targeted drug administration strategy once clusters are identified. Genomic clustering was not correlated with human host-specific factors, such as age, sex or treatment compliance. These variables certainly influence infection levels from an epidemiological perspective, but not to the degree of genetic relatedness between worms in this community. While this study has highlighted the focal pattern of infection within communities, the factors and reasoning behind the focality need further exploration in future work. Agricultural practices are likely to significantly affect these foci of infection, with the traditional practice of human fertiliser in farming maintaining STH transmission across rural communities in endemic regions[48–50]. Furthermore, agricultural practices such as livestock ownership add additional reservoirs of infection risk to treated communities, given the close evolutionary relationship and host sharing of *A suum* and *A*

*lumbricoides*[27,29]. Whilst the ownership of pigs is not prevalent in Wolaita[51], it is possible that zoonotic reservoirs of infection play a role in creating human transmission foci[26,27,52,53]. Further investigation within these spatially distinct clusters into WaSH and compliance status is required to unravel the factors influencing the continued presence of specific transmission foci.

At the community-level, we found positive genome-wide Tajima's D disaggregated across individuals of varying drug compliance designation. The discovery of positive Tajima's D values and an increased proportion of intermediate-frequency alleles compared to number of rare alleles (Supplementary Fig. 5) indicates a population under contraction[54]. A shrinking population of *A. lumbricoides* may be indicative of the cMDA treatment history of the region. Biometric tracking of participants has allowed the programme to collect individual-level data on cMDA compliance and infection status. While worm expulsions indicated that partially and non-compliant individuals harbour a much larger fraction of the total parasite population (Supplementary Data 1) overall, these worms are part of a shrinking population due to the years of MDA treatment in the community. This study is the first to utilise individual-level drug compliance data alongside population genomic analysis to highlight the biological factors that underpin how non-treated individuals act as an infection reservoir following cessation of expanded treatment programmes. For control programmes to achieve an elimination goal, further understanding of how MDA is affecting the population size of infective worms in communities alongside traditional monitoring and evaluation processes is crucial[55]. As the sensitivity of traditional epidemiological data collection diminishes in lower prevalence communities, understanding the true worm burden of individuals across a repeatedly treated village is vital to understand when thresholds have been met, and cessation of drug treatment intervention is possible[56]. Developing a framework around worm expulsions and genomic analysis offers promise in this regard, as has been shown here, however, this process will need significant upscaling to offer significant power to *Ne* values that are robust enough to offer control programmes monitoring and evaluation

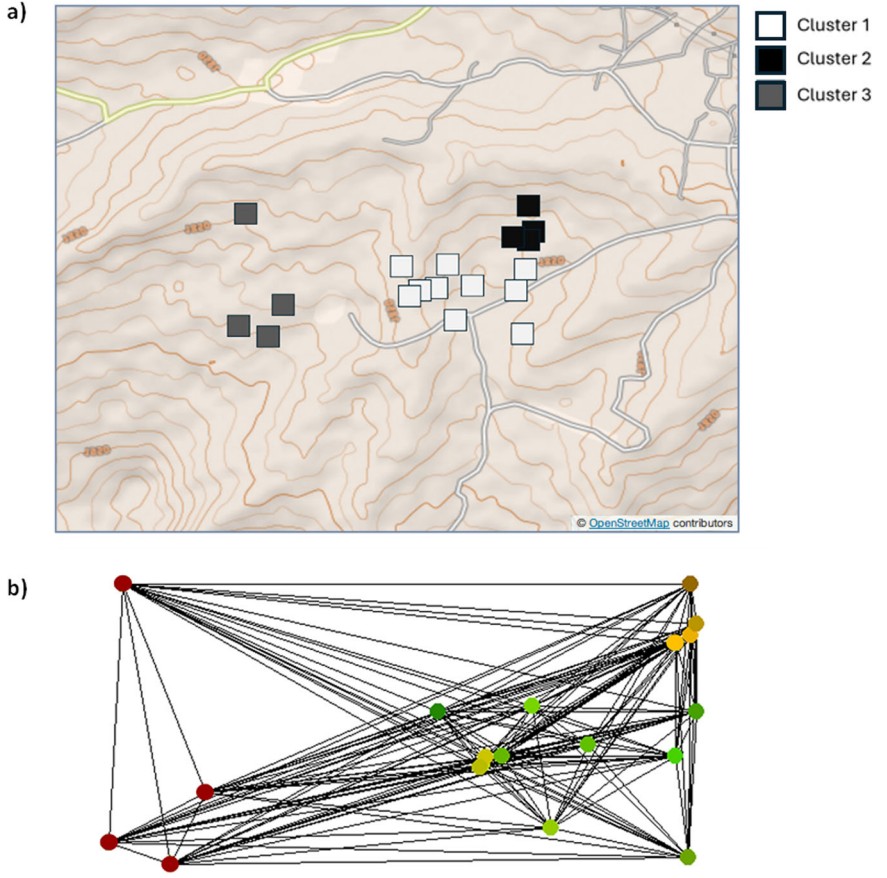

**Fig. 6 | Spatial clustering model of *Ascaris lumbricoides* within a single community. a** Represents the first global axis scores from full landscape sPCA ($n = 54$). Each georeferenced household is depicted with a square, where the dark grey colour represents the major spatial genetic clusters of sampled households. **b** Network used to define spatial weightings for sPCA analysis, colours indicate the spatial genetic cluster assignments. Map tile © OpenStreetMap contributors, licensed under the Open Database License (ODbL).

benchmarks. To quantify the relationship between demographic population size fluctuations, infection intensity and drug compliance, further work needs to be performed under differing treatment regimens such as the WHO-prescribed sMDA and expanded cMDA approaches.

In recent years, evidence of reduced efficacy of albendazole has arisen in some settings with endemic infection and regular MDA[43,57]. The nature of mutations that would be classified as rare resistance-associated variants at a single locus to be swept to fixation within a population would be classified as a hard selective sweep. These would leave a strong selective signature in the genome around the locus, which can be detected by several methods such as the $F_{ST}$ and haplotype tests (iHS) that have been employed in this study. Other scenarios, such as the presence of a resistance allele across a range of genetic backgrounds, which subsequently would rise to gradually elevated frequencies (soft sweep), would be harder to detect and may have been missed in this study. This type of soft sweep would be more likely to be detected across a multi-community study within a larger geographic region. If variants are associated with a range of genetic backgrounds, this would be demonstrated through continued genetic variation across a longitudinal study, which has yet to be performed in STH species. The expansion of MDA programmes across endemic regions over the past decade highlights the need to monitor the possibility that resistance may develop and establish rapidly across worm populations[44]. Resistance to benzimidazole anti-helminthic drugs in other parasitic nematodes has identified causal SNPs within the beta-tubulin gene[41,42]. Whilst the gene has yet to be implicitly implicated in reduced albendazole efficacy in treated populations of *A. lumbricoides*,

a small number of studies have investigated the presence of resistance-associated mutations in *A. lumbricoides* collected as part of control programmes, finding them occurring at low frequency (0.05%)[58]. Alternatively, geographically disparate studies in Kenya, Haiti and Panama indicated a high proportion of the F167Y (position 200) resistance-associated mutation[33]. Worms collected as part of this study did not display any specific substitutions related to albendazole resistance in line with global screens for the variant[59]. The study community showed a high proportion of infected individuals being designated as 'Lost infection' in subsequent annual surveys following accepted treatment (Table 1). Therefore, albendazole still shows good efficacy across this community. There were study design limitations with respect to our insights into drug resistance in STH. Firstly, the worms that were sampled in this study were collected via chemo-expulsion, with dead adult worms expelled within the stool of infected individuals. Therefore, all worms were susceptible to the albendazole treatment delivered in this study. For this reason, the work described here should be considered as an appraisal of the genome-wide selection landscape across *A. lumbricoides* worms, which have undergone multiple rounds of treatment whilst maintaining transmission. Our methods implicate regions under selection and the potential functional ramifications of loci within them. However, under a sampling strategy that contrasts resistant and susceptible worms, the bioinformatic workflow described here shows promise in detecting loci involved in drug resistance or increased transmissibility. To establish a framework to assess genetic drug resistance in STH, structured longitudinal collections should be employed alongside a well-delivered control programme. Sample collection at baseline, yearly and endline

surveys within a control programme would allow assessment of genome-wide patterns of selection correlated with MDA delivery. Additionally, establishing biobanks of high-quality tissue samples will allow us to robustly compare worms that show reduced clearance in future control efforts with historical samples. This would permit the detection of selection in response to drug[60].

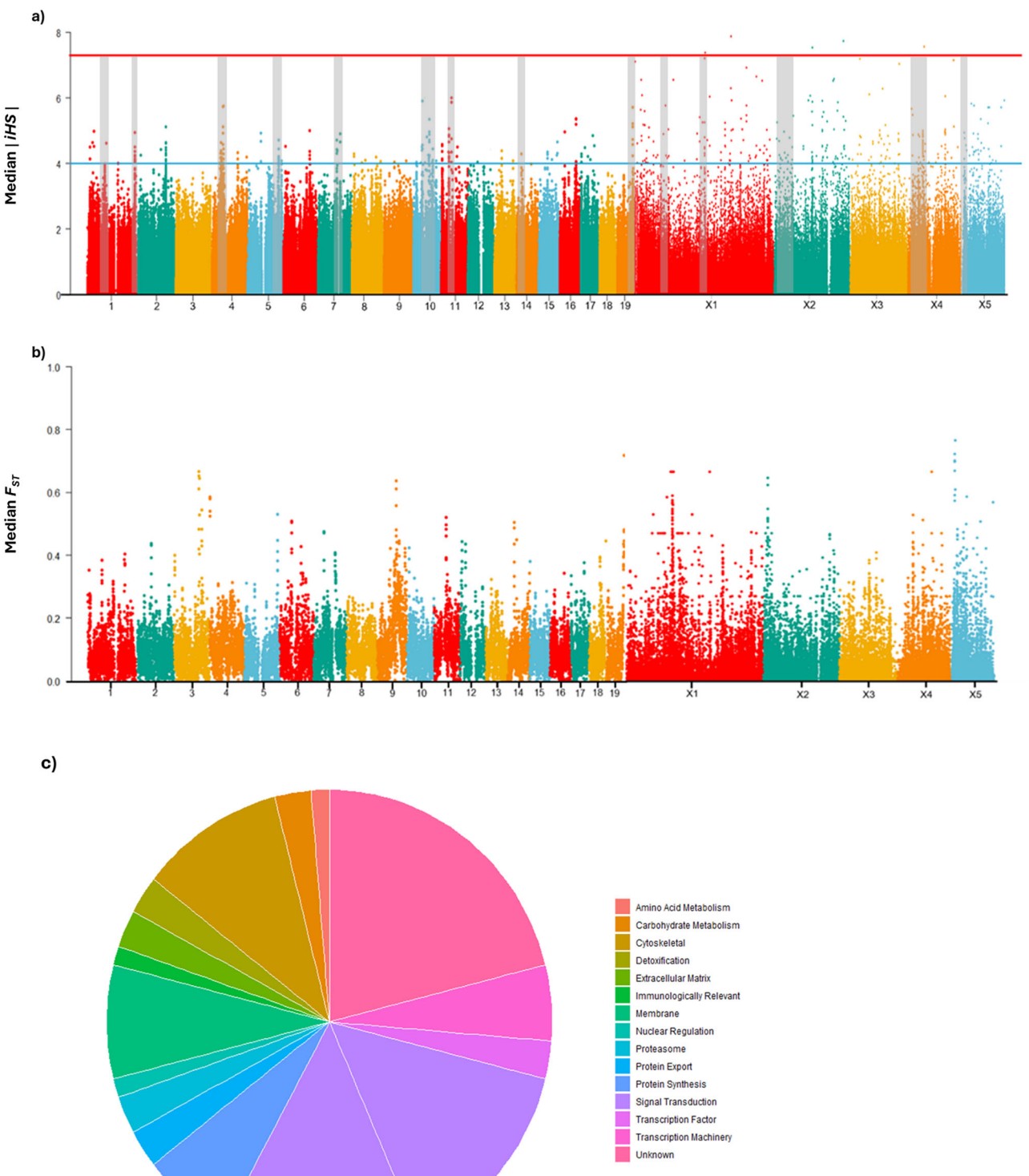

**Fig. 7 | Genome-wide integrated haplotype scores (iHS) within the Korke Doge population of expelled worms (54). a** Regions that show increased iHS scores in conjunction with elevated FST are highlighted within the grey-filled boxes. **b** Fixation index FST between fully compliant individuals who have accepted the drug each round of treatment, and non-compliant individuals who have accepted treatment at only the latest round of treatment within Korke Doge. Windows with a 300 kb of each other were joined into a continuous region of selection, regions with 8 windows with elevated values across both iHS and FST scores were discarded. **c** Pie chart of gene functions for genes in candidate regions of positive selection.

The main conclusions of the study are twofold. First, the study of one *A. lumbricoides* population showed fine-scale spatial division across households and sets of households that have received repeated cMDA. Transmission seems to be very household-focused and the genetic distance between worms increased along with the spatial distance between households. Second, at present, despite long-term repeated treatment, there appears to be no indication of resistance to albendazole treatment in this parasite population. However, we did identify several genomic regions with strong signals of selection that could indicate gradual adaptation to long-term albendazole exposure or other selective pressures. Further, much larger-scale genomic surveillance across endemic regions, particularly alongside control programmes utilising tracked cohorts, is clearly desirable.

In low-prevalence communities, the key policy question at present is how best to focus control efforts. Improved WaSH is essential in terms of its broad-scale impact on many infections. Continued prophylactic cMDA of large populations, when only a few are infected, is less clear-cut. MDA alone at very high coverage repeatedly for many years can stop STH transmission, but the expense and the continued availability of drug donations in resource-poor settings are less certain. Moreover, this repeated use of single drug treatment on mass-scale without understanding methods to monitor specific loci involved in drug resistance, will put future programmes at risk[44]. Targeting treatment to the few who remain infected, many of whom are predisposed to infection, is one possibility as this study suggest that the household is the basic unit of transmission for *A lumbricoides*. However, the use of mass diagnosis by parasitological stool examination or other more sensitive methods[61] is also expensive, certainly if repeated for all MDA rounds. Predisposition and the identification of household hotspots of transmission could significantly reduce the workload and associated expenses, since once identified in low-prevalence settings, treatment would in the future just focus on these households. Molecular approaches are widely used in the study of the epidemiology and control of viral[62] and bacterial[63] pathogens, but are very rare in the study of the neglected tropical diseases. The costs of sequencing large genomes continue to decline, and hence larger-scale studies of 'who infects whom' and routine genetic surveillance will become more feasible. It is hoped that molecular epidemiological approaches within a framework performed here will be employed more widely to further the understanding of transmission in low-prevalence settings and give insights into how to stop it.

## Methods

### Worm collection and ethical approval

This study was performed with approval from the Ethiopian Public Health Institute Institutional Review Board (EPHI-IRB: Protocol number EPHI-IRB-471-2022, Minutes No. 110). This included the collection and publication of indirect identifiers and analysis by researchers at Imperial College, London and SRUC, including epidemiological metadata and anonymized household GPS location. Within the Korke Doge research community, sample collection was led through zonal health workers and community health extension workers (HEW) with the compliance of members of the Ethiopian Public Health Institute. Village leaders were contacted for consent to the study through zonal health workers before the study took place. Where the collection of samples was from a child under the age of 18, parents'/guardian' permission was sought. Any child included in the study was asked to sign and give informed consent after receiving full information about the study. Participation was voluntary, with children and participants being able to withdraw from the study at any time. Access to treatment was not dependent on consenting to participate in the study. All individuals in the community were given access to treatment with albendazole. The worms collected were expelled through treatment delivered by the Ethiopian Federal Ministry of Health and did not require additional ethical approval to work with animals. Whole worms

are housed long term under refrigerated (−20 °C) conditions at the Ethiopian Institute for Public Health.

Worm expulsion and collection took place following four rounds of cMDA in the Korke Doge community (Fig. 1). Worms were collected from 20 households within two Gotts (collection of houses within a community) in the Korke Doge (Fig. 2) kebele (a collection of household clusters equivalent to a small village) after a routine deworming activity implemented by the Geshiyaro project. This was under an expanded, community-wide framework whereby all the community and pre-SAC in the MDA eligible population (Albendazole ≥1 year) received treatment. The MDA coverage has been scaled from the 2019 baseline study to reach a verified 90% coverage, with a 10% confidence interval in all targeted groups on a biannual basis[11,64]. Following the treatment of 400 mg of albendazole delivered household-to-household, worms were collected from the stool of individuals for 5 days. Korke Doge kebele was chosen for a sample site due to the in-depth epidemiological understanding of infection, prevalence and drug compliance of the cohort within the community[35,65–67], which enhances the ability to interpret the outputs from population genomic analysis. Worm expulsion occurs over 5 days following the ingestion of Albendazole[68] and hence entails the collection of stool and its examination for worms. One hundred individuals were chosen to sample, and stool was examined daily for 5 days following treatment. These 100 individuals were chosen following previous positive results for infection occurring across the past annual sampling of the study cohort. To establish the spatial household-to-household transmission dynamics across the community, sampling was structured to include only households where worms were collected from multiple members within a GPS-linked household. Following the removal of a worm from an infected individual's stool, whole worms were isolated, washed in molecular grade ethanol to remove stool contamination from the cuticle and stored within 15 ml microcentrifuge tubes. A single worm was selected from each infected individual per household to establish a dataset for comparison of genetic diversity indices within and between households. Where multiple worms were available, the storage tubes containing whole worm tissue and 98% ethanol were stored in 4 °C cool boxes while on site before transport to −20 °C freezer within 2 h of collection in a hospital clinic in Sodo, in the Wolaita zone. Whole worm tissue was transported on dry ice to SRUC Inverness and stored at −20 °C before DNA extraction within a week of collection.

### Epidemiological data collection

Annual parasitological mapping within the Geshiyaro project took place across 45 sentinel sites in 45 kebeles, whereby stool samples were collected, and participants answered a WaSH infrastructure access questionnaire. A demographically balanced sample of 150 participants were selected per site, consisting of 15 individuals of each sex in each age group: pre-SAC (aged 1–4 years), SAC (aged 5–14 years), adolescents (aged 15–20 years), young adults (aged 21–35 years) and adults (aged 36–100 years). All participants were enroled in the project using electronic data capture fortified with biometric fingerprint technology for the registration and subsequent identification of participants. This generated an 11-digit participant number, which could be identified either biometrically, by scanning their ID card, or by searching for their name stored within records from previous cohort. Where sentinel sites existed in census districts (*n* = 16 kebeles), participants could be linked to their census record and cMDA record.

All faecal samples collected from a participant were provided with a six-digit sample number identifiable through scanning a QR code. This code was placed on sample pots and Kato Katz diagnostic slides to reduce human errors in the labelling of samples. This also allowed for anonymised diagnostics in the laboratory. Quadruple Kato Katz slides were analysed, generated from two slides prepared per sample over two consecutive days. A minimum of 6 months was allowed between cMDA distribution and sentinel site parasitological monitoring to

ensure any recrudescence (parasite reinfection) could be captured, signifying continued STH transmission.

Anthelmintics were distributed by HEW, a network of care workers stationed in each kebele across Ethiopia, underpinning the national healthcare system. A single dose of albendazole was offered to members of the community aged one year old and above, with syrup distributed to infants aged one to four years old. From Year 4 onwards, albendazole was offered biannually.

During cMDA, each participant was identified using their biometric fingerprint or ID card, which linked their 11-digit participant number to their treatment behaviour. Each participant was therefore categorised as either 'missed' or 'contacted' by a HEW, with the latter further categorised as 'accepting' or 'refusing' the offered drug, and subsequently 'refusing' or 'swallowing' the accepted drug, as recorded by the HEW. The eligible population for albendazole was calculated as the population aged 1 year old and above, excluding all pregnant women.

Prevalence of *A. lumbricoides* infection was calculated from four Kato Katz slides for each individual, and 95% confidence limits were calculated using positive binomial confidence limits. Intensity of infection was calculated from the averaged mean egg count across the four Kato Katz slides multiplied by 24 to generate the mean eggs per gram. Negative binomial confidence limits were calculated for egg count intensity measures[52].

## Treatment compliance datasets

During cMDA, each participant was identified using their biometric fingerprint or ID card, which linked their 11-digit participant number to their treatment behaviour. Each participant was categorised as either 'missed' or 'contacted', with the latter further categorised as 'accepted' or 'refused' the offered drug, and subsequently 'refused' or 'swallowed' the 'accepted' drug, as recorded by the HEW. The eligible population for albendazole was calculated as the population aged 1 year old and above, excluding all pregnant women. The term 'fully compliant' defines individuals who have been categorised as 'accepted' and 'swallowed' albendazole at each round, whilst the term 'non-compliant' refers to individuals who 'refused' at each round of treatment. The term 'semi-compliant' defines individuals who 'accepted' albendazole at some rounds of MDA, but not all.

## DNA extraction and sequencing

DNA extraction was performed on snip sections of worm tissue across 54 adult worm samples. Approximately 15 mg of tissue was cut from each worm (8 mm$^3$). Female worms were selected due to size and amount of tissue. Precautions were taken post-collection, with the possibility that female *Ascaris* might be containing fertilised eggs, which can cause spurious genotypes[69]. The uteruses of female *Ascaris* were removed to prevent contamination by male sperm during DNA extraction. DNA was extracted from somatic tissue (body wall or intestine) following the removal of the cuticle. Previous work indicates that there is no significant variance in DNA yield in comparison of intestinal or musculature tissue[27,70]. DNA extraction was performed using the Qiagen MagAttract (Qiagen: cat.no 67563) magnetic beads extraction methods across all worms according to the manufacturer's workflows. DNA fragmentation was assessed through gel electrophoresis, with further quantification performed using Qubit™ 1X dsDNA High Sensitivity quantification kit (ThermoFisher: cat.no Q33230).

Paired-End Genome Libraries of *A. lumbricoides* DNA samples were sequenced using 150 bp Illumina HiSeq 2500 (www.illumina.com) short-read paired-end sequencing. DNA was quantified by UV Spec and Quant-iT™ PicoGreen™ (ThermoFisher: cat.no P7589). A 100 ng of DNA based on picogreen quantification was used as template for NGS library preparation using the TruSeq Nano DNA Sample library prep kit (Illumina: cat.no FC-121-4001). Primer-dimers in the libraries were removed by additional AMPure beads purification (Beckman Coulter

cat.no A63881). Sequencing was performed to obtain a minimum genomic depth of 20× coverage for each sample. Mate-Pair Genome Libraries—Two samples were selected for mate-pair sequencing, based on the quality of the DNA preparation. The mate-pair libraries were generated using the Nextera Mate Pair Library Prep Kit (Illumina cat.no FC-132-1001), following the gel-free method with the only modification that M-270 Streptavidin binding beads (ThermoFisher cat.no 88817) were used instead of M-280 beads. The libraries were amplified for 15 cycles, given the low DNA input going into the circularization phase. The mate-pair fragment size averaged 6 kb with a range of 2–10 kb fragments.

## Variant discovery and annotation

Raw sequence reads from all 54 samples were trimmed using FastQC to remove low-quality bases and adaptor sequences. Trimmed sequence reads were aligned using BWA mem (v.0.7.17)[71]. Owing to the 99% similarity in terms of structure and content between the *A. lumbricoides* and *A. suum* genome reads were aligned to the chromosome level, higher resolution *Ascaris suum* genome (accession number: ASM1343314v1)[37]. PCR Duplicates were marked using PicardTools (v.3.3.0) MarkDuplicates[56]. Variant calling was performed using GATK HaplotypeCaller (GATK v.4.0) in gVCF mode[72], retaining both variant and invariant sites. The individual gVCFs were merged using GATK CombineGVCFs, and joint-call cohort genotyping was performed using GATK GenotypeGVCF's. Variant sites with SNPs were separated from indels and mixed sites (variant sites having both SNPs and indels) using GATK SelectVariants. GATK VariantFiltration was used to filter both these independently. SNPs were retained when meeting the following criteria in the filtering process: QD ≥ 2.0, FS ≤ 60.0, MQ ≥ 40.0, MQRankSum ≥ −12.5, ReadPosRankSum ≥ −8.0, SOR ≤ 3.0. Additional filtering, including the variant sites that contained indels or mixed sites, was also retained when meeting the following criteria: QD ≥ 2.0, FS ≤ 200.0, ReadPosRankSum ≥ −20.0, SOR ≤ 10.0.

VCFtools (v.0.1.16) was used to exclude accessions with a high rate of variant site missingness (>5% of sites calling missing genotype), subsequently removing sites where >10% of accessions had a missing genotype. The filtering process outlined above formed the primary VCF file for downstream analysis. To analyse nucleotide diversity (Pi) and fixation index ($F_{ST}$), a second VCF file was produced. We used the VCFtools to filter both variant and invariant sites with >80% missing variants, a minimum mean read depth of 5 and a maximum read depth of 50. Variant sites that were found to be significantly out of Hardy-Weinberg equilibrium ($p < 0.001$). Functional annotation of SNPs and indels in the primary analysis VCF file was performed using SnpEff (v.5.0e) with gene annotations downloaded from WormBase ParaSite.

## Population genomic structure and diversity

SNPs that exhibit strong linkage disequilibrium are highly correlated, meaning they provide similar information regarding population differentiation, and confound the results of statistical analysis. To establish population structure across a closely related sample size we removed all variants found to be in strong linkage disequilibrium using PLINK (v.2.0). PCA was performed on the remaining 1,843,016 autosomal SNP using PLINK (v.2.0)[73], *K* values (hypothetical ancestral populations) were established, ranging from 1 to 20 (Supplementary Fig. 3).

The 1,843,016 autosomal variants were used to construct a neighbour-joining tree. An identity-by-state distance matrix was created with distances expressed as genomic proportions generated by PLINK (v.2.0). The ape bionj algorithm[74] was performed on the resulting matrix and visualised using ggtree[75]. PIXY (v.0.95.02)[76] was used to calculate the nucleotide diversity ($\pi$), fixation index ($F_{ST}$) and absolute divergence ($d_{XY}$), in 5 kb sliding windows with no-overlap across autosomes for each household and designated treatment compliance dataset.

## Spatial population genomics

To understand the extent of variation in genetic structure by location and identify spatial genetic discontinuities across the study community, several related methods were employed. First, a subset of 1,000,000 autosomal variants was created from a VCF file using the GATK SelectVariants function (GATK v.4.0)[77]. Bayesian clustering with BAPS (v5.3)[78] was used to test for the presence of multiple genetic populations under Hardy-Weinberg equilibrium. PCA was employed to identify major trends in the genomic structuring through ordination. The fineSTRUCTURE method was used to assess genetic structuring through differences in shared co-ancestry[79]. Spatial principal components analysis was employed to assess the genetic structuring by capturing patterns via spatial autocorrelation within the spatial genetic variability[80,81].

## Detection of genome-wide signatures of selection

Statistical phasing was performed using Beagle (v.5.0). Bi-allelic variants were then used to calculate the iHS using Selscan (v.1.2.0a)[82] for each chromosome separately. Normalisation was performed for all Selscan outputs across all chromosomes using norm (v.1.2.0a) distributed with Selscan. Median iHS and XP-EHH scores were calculated for 2 kb non-overlapping windows along each chromosome. VCFtools was used to calculate fixation index ($F_{ST}$) and $\pi$ in 2 kb sliding, non-overlapping windows across each chromosome. The ratio in nucleotide diversity was calculated as the median $\pi$ value within each 2 kb window. Consistent signals across multiple statistics were taken as strong evidence of selection[83]. The highest 0.25% of iHS and $F_{ST}$ windows were considered candidate regions of selection. As additional supporting evidence, we calculated Tajima's D statistic in 2 kb non-overlapping windows using VCFtools. As above, the same windows with fewer than 20 variants per 2 kb were removed. In addition, per variant site $F_{ST}$ and $\pi$ values were calculated using VCFtools.

## Reporting summary

Further information on research design is available in the Nature Portfolio Reporting Summary linked to this article.

## Data availability

The sequencing data generated in this study have been deposited in the NCBI project repository PRJNA1195031 of NCBI GenBank (https://www.ncbi.nlm.nih.gov/sra/). Individual sample accessions are listed in Supplementary Data 2. Genome and annotation files are available through WormBase Parasite (http://parasite.wormbase.org/Ascaris_suum_prjna80881/Info/Index/). Owing to the personal data that is contained regarding the cohorts used in this study, names have been removed from the epidemiological data. Additionally, due to the data here being derived from Ethiopian nationals, longitudinal individually-linked epidemiological data is owned by the Ethiopian government and can be shared at request with the corresponding author and/or the Ethiopian Institute of Public Health (https://ephi.gov.et). Source data are provided with this paper.

## Code availability

The source codes and bioinformatic workflow of genomic data handling, analysis and visualisation can be found open-access via the corresponding author's GitHub account—(https://github.com//TobyLand/Ascaris_PopGen). This is available via Zenodo: https://doi.org/10.5281/zenodo.15124860.

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

## Acknowledgements

We thank the technicians, local health workers from the woreda health division in Wolaita. We would also like to thank the technicians involved in the collection of samples in Korke Doge kebele: Selamawit Ginjo, Silas Dea, Moges Mekonin, Selamu Dana, Simegn Mengistu, Abraham Anjulo, Michael Tumato and Misgana Zekarias. We would also like to thank the community leaders and members of the Korke Doge kebele who were recruited into the study, and their cooperation with everyone involved. Research for this paper and the authors was funded by The Children's Investment Fund Foundation ('CIFF'), UK, through a grant to the London Centre for Neglected Tropical Disease Research ('LCNTDR') at Imperial College London, UK and SRUC. The views, opinions, assumptions or any other information set out in this study are solely those of the authors and should not be attributed to CIFF or any person connected with CIFF. The grant code attributed to the funding of authors is R-1805-02741 (https://ciff.org/grant-portfolio/geshiyaro/). This work was performed by: T.L.

## Author contributions

T.L., R.M. and R.A. conceived of the project, for which R.A. secured funding. T.L., R.M., K.F., R.S.G., E.F.L., U.A., G.T., B.M. and R.A. planned and coordinated the fieldwork, which was carried out by R.M., R.S.G., T.L. and M.B. T.L. and M.B. planned and coordinated the molecular biology and sequencing. T.L. and J.H. analysed data with input from J.E. and S.P.L., and T.L., R.M., R.A. and J.H. wrote the paper with input and approval from all authors.

## Competing interests

The authors declare no competing interests.
