## [Transparent Peer Review file · Nature Communications]

Molecular epidemiology of *Ascaris lumbricoides* following multiple rounds of community-wide treatment

Corresponding Author: Dr Toby Landeryou

Version 0:

Reviewer comments:

Reviewer #1

(Remarks to the Author)

The manuscript entitled Molecular epidemiology of *Ascaris lumbricoides* following multiple rounds of community-wide treatment: who infects whom provides a novel molecular epidemiological approach to help to understand the remaining hotspots of low prevalence *A. lumbricoides* infection in communities. The work is concise, clear and well structured and will provide a useful template for other investigators who face similar challenges when making decisions after several rounds of mass drug administration to eliminate this helminth. The work is well referenced and the figures and tables are useful and nicely augmented by the supplemental tables. The methodology is sound and the conclusions well presented.

The manuscript although well written is littered with numerous typographical errors, which need to be addressed as follows:

- Line 65 – full stop missing
- Line 69 – renainging – should it be remaining?
- Line 70 rewrite this section
- Line 125 add inverted comma after accepted
- Line 126 - Is referring to referred
- Line 154 – change comma to full stop
- Line 163 – change comma to full stop
- Line 206 – remove question mark
- Line 209 – insert the before community
- Line 217 – insert s into population
- Line 245 – removed from compares
- Line 249 – remove full stop in figure and add comma
- Line 271 – remove bold from Table three
- Line 292/3 – change sustain to maintain
- Line 297 – close bracket
- Line 299 – change secondly to second
- Line 299 – change finally to third
- Line 310 – replace spatially with spatial
- Line 311 – Capitalise genomic
- Line 318-322 – reword to add clarity

Reviewer #2

(Remarks to the Author)

In the study entitled “Molecular epidemiology of *Ascaris lumbricoides* following multiple rounds of community-wide treatment: who infects whom?”, the authors sequenced 54 whole-genomes from individuals in a longitudinal cohort epidemiological study of transmission and drug treatment extending over 6 years. This study provided valuable insights for shaping future control policies in low prevalence settings, highlighting the importance of more targeted treatment of infection hotspots.

However, there are some points that the authors need to address to make the manuscript clearer:

1. Detailed explanation of “who infects whom”.

2. The citation format is not standardized.
3. References 26 and 30 are identical.

Reviewer #3

(Remarks to the Author)

Nature Communications Review Jan 2025

This study describes the generation of whole genome sequence data from adult *Ascaris* worms obtained from people living in one community in Ethiopia enrolled in a longitudinal study investigating interventions to interrupt the transmission of soil-transmitted helminths and schistosome infections. By the time of worm collection, participants had received multiple rounds of mass drug administration over a 6-year period. The study links genomic and spatial data at a community level. It also explores regions of positive selection across the *A. lumbricoides* genome.

This is a very interesting study generating important new data and the analyses are carefully conducted. However, there are a couple of limitations which should be highlighted. Firstly, the worms which were analysed were obtained after standard MDA treatment with albendazole. This means that the sampled worms were sensitive to albendazole. There may well be other worms with reduced response to albendazole circulating in the population which would not be sampled using this approach. This limitation should be mentioned in the Discussion and the results interpreted accordingly. Secondly, no samples were collected at the beginning of the study for comparison with samples collected after 6 years of MDA. This should be mentioned as a limitation in the Discussion.

Other recommended revisions are as follows:

Abstract

Line 24 – insert “*Ascaris* worms obtained from” after “from”

Line 30 – should “no high” be “low”?

Introduction

Lines 35-39 – There are more recent estimates of the DALYs associated with intestinal nematode infections and number of people infected with *Ascaris* from the Global Burden of Diseases study, see the Institute for Health Metrics and Evaluation. Please update the figures accordingly.

Line 41 – replace “of” with “using”

Line 46 – perhaps “pre-SAC” is better than “pre-“

Line 50 – clarify “To achieve transmission, break a 2% prevalence must be achieved”

Line 59 – perhaps “is seen” or “exists” is better than “perturbs”

Line 69 – replace “reanaiging” with “remaining”

Line 70 – replace “result due to many by” with “may be attributed to”

Line 101- states “largest WGS study”, but Easton et al. 2020 paper performed more (n=68, I believe, albeit maybe at a lower coverage)

Materials and methods

There is no description of the statistical methods used to analyse the epidemiological data. These should be included.

It is not clear where the genomic data generated during this study will be made available to the scientific community. Details should be provided.

Most of the description of the bioinformatics methods is in the supplementary material. It would be useful to include more in the main body of the text, and signpost the supplementary material where needed,

Will any of the scripts used be available on Git-hub or an alternative repository? Line 106-107 – why were 20 households selected for worm collection? Why was Korke Doge kebele chosen for this study?

Line 109-110 – mention that 400 mg of albendazole was delivered as part of the community-wide MDA administered during the Geshiyaro project. Mention how many individuals had their stool examined for 5 days to collect worms and how these individuals were selected. Also state how worms were processed after collection. Were the worms washed? How? How were the worms stored and transported?

Line 110 – “Epidemiological data collection” should be on a new line

Line 116 – should it be seven rounds rather than four rounds of cMDA? A diagram summarising when the cMDA took place over the six-year period would be helpful.

Line 129 – Where did DNA extraction take place? How long was the gap between worm collection and DNA extraction? Were the worms pre-washed/rinsed in any buffer or retained in any storage media? How large were the snip sections? What tissue was used for DNA extraction and this just somatic tissue or also germline? Was the cuticle removed before DNA extraction? How was decided which adult worms to sequence? It was mentioned in the supplementary material that DNA was extracted from 66 worms but 54 are mentioned in the main text. Please correct this discrepancy. Were any of the collected worms sexed, particularly the worms that were sent for sequencing?

Line 130 – Was Qiagen MagAttract kit used according to manufacturer’s instructions? If so this should be mentioned. Otherwise, modifications should be described.

Line 132 – Apart from DNA quantification, was there any quality control to look at level of DNA fragmentation e.g. by TapeStation?

Line 140 – “reads were aligned using BWA mem” could clarify here what they were aligned against (include accession numbers)

Line 145 - The use of GATK VariantFiltration for filtering SNPs and indels is mentioned, but there is no information on specific filtering thresholds (e.g., minimum QUAL score or min depth).

Line 152 – Explain why variants that were in strong linkage disequilibrium were removed before conducting population genomic analysis.

Line 152 – Add abbreviation for principal component analysis (PCA)
Line 153 – PLINK – missing the version number
Line 154 – “1 to 20,” to “1 to 20.”
Line 154 – should it be ADMIXTURE? If so, please include the version where possible
Line 157 – PIXY – version number missing
Line 164 – change “principal component analysis” to PCA
Line 166 - change “principal component analysis” to PCA

Results

Line 170-175 – It could be clearer that prevalence was determined using faecal egg counts (mention the method used) vs identifying worms in faeces.
Line 172 – suggest using “collection” rather than “expulsion” as worms were collected after MDA rather than after using a treatment to stimulate expulsion.
Line 194 – the reference *A. lumbricoides* genome was used for alignment. Was any consideration given to using the reference *A. suum* genome which is higher quality and closely-related?
Line 198 – Is the coverage range correct? The lower figure does not appear to match that in the supplementary data.
Line 198 – Did you check to see if low read depth was consistent across the samples/ focused on specific loci? Line 202 – “sample” should be “samples”
Line 203 – use PCA abbreviation only
Line 206 – remove question mark (sentence could be improved for clarity)
Line 210 – should “establish” be “established”? Line 211 – insert space after “(Supplementary Fig 4).”
Line 212 – how was the subset of autosomal variants created?
Line 217 - Should “population” be “populations”?
Line 228 – “Adolescents” should be “adolescents”
Line 229 – “Adults” should be “adults”
Line 245 - “highlighted reduced geneflow between worms from compared from within households compared to between” - revise for clarity
Line 278 – Remove “Within these regions we identified a single locus containing beta tubulin gene.” This repeats earlier text.
Line 280-282 – There is quite a bit of discussion in the literature as to whether beta-tubulins are linked to reduced efficacy in *A. lumbricoides*. The link is clear in clade 5 nematodes infecting ruminants and horses but there is no clear evidence in ascarids as of yet. This warrants further discussion but this is probably better placed in the Discussion section rather than the Results.

Discussion

Line 311 – change “genomic” to “Genomic”
Line 316 – Please clarify how the data indicates a shrinking population of *A. lumbricoides* across Korke Doge.
Line 322 – What about environmental reservoirs?
Line 324-325 – Sentence could be made clearer
Line 331 – change “If variants associated” to “If variants are associated”
Line 349 – I think somewhere it needs to be mentioned that in other communities using faeces as fertiliser is a major contributor human transmission. Therefore, more molecular studies of this nature will be required to actively support control initiatives. You could mention pig-human transmission in this regard, also.
Figure 1 – Could include a scale
Figure 2. - Include total numbers of people tested.
Figure 2 – “a” is bold but “b” is not.
Figure 2 - “The target 2% prevalence of *A. lumbricoides* is shown as a grey dashed line”. This is not very visible
Figure 2 – You could include the WHO guidelines of ≤ 4999 epg
Table 1. Include total numbers for each time period.
Figure 3 – Reword “between the 54 *A. lumbricoides* produced sampled for WGS”
Figure 6 – higher resolution image required

Supplementary text

Should supplementary text be referenced in the main text?
DNA extraction and sequencing – Change “snips” to snip. Was the Invitrogen Qubit high sensitivity or broad-spectrum assay used? Ensure *A. lumbricoides* is italicized correctly throughout. Was 100bp or 150bp used for Illumina sequencing?
I understand that this is the supplementary text, but include version numbers of software where possible.
“five years of treatment” mentioned at the top of page 2. Is it five or six years of treatment?
The supplementary text mentions that DNA was extracted from 66 worms and that 68 DNA samples were sequenced. This contrasts with 54 worms in the main text.
In “Spatial population genomics” paragraph, the first sentence is not a proper sentence. There is an extra full stop at the end of the paragraph.
Formatting of Supplementary Figure 4 could be improved to improve readability.
Supplementary Figure 5 – Check this sentence: “All three plots taken in the round indicate that three genetic clusters exist in three genetic clusters”. Also “e plot represented” should be “e plot represents”.

(Remarks to the Author)

Version 1:

Reviewer comments:

Reviewer #3

(Remarks to the Author)

The authors have addressed nearly all our comments and the manuscript is significantly improved.

A couple of minor issues

There are a few sentences that are confusing i.e line 219, so please read carefully.

Check the figure numbers as the legend jumps from 5 to 7a.

(Remarks on code availability)

To all Reviewers,

First and foremost I would like to thank each of the reviewers in regards to agreeing to review this document for publication in Nature Communications. I have amended the manuscript according to your comments and where relevant included the specific lines in the manuscript where this edit has taken place

Thank you

Reviewer #1 (Remarks to the Author):

The manuscript entitled Molecular epidemiology of *Ascaris lumbricoides* following multiple rounds of community-wide treatment: who infects whom provides a novel molecular epidemiological approach to help to understand the remaining hotspots of low prevalence *A. lumbricoides* infection in communities. The work is concise, clear and well structured and will provide a useful template for other investigators who face similar challenges when making decisions after several rounds of mass drug administration to eliminate this helminth. The work is well referenced and the figures and tables are useful and nicely augmented by the supplemental tables. The methodology is sound and the conclusions well presented.

The manuscript although well written is littered with numerous typographical errors, which need to be addressed as follows:

- Line 65 – full stop missing - **amended**
- Line 69 – renainging – should it be remaining? – **amended – See line 76**
- Line 70 rewrite this section – **amended – see line 78-79**
- Line 125 add inverted comma after accepted - **amended**
- Line 126 - Is referring to referred – **amended – see line 176**
- Line 154 – change comma to full stop - **amended**
- Line 163 – change comma to full stop - **amended**
- Line 206 – remove question mark - **amended**
- Line 209 – insert the before community – **amended – see line 283**
- Line 217 – insert s into population - **amended**
- Line 245 – removed from compares - **amended**
- Line 249 – remove full stop in figure and add comma - **amended**
- Line 271 – remove bold from Table three - **amended**

- Line 292/3 – change sustain to maintain - **amended**
- Line 297 – close bracket - **amended**
- Line 299 – change secondly to second - **amended**
- Line 299 – change finally to third - **amended**
- Line 310 – replace spatially with spatial - **amended**
- Line 311 – Capitalise genomic - **amended**
- Line 318-322 – reword to add clarity – **amended – see line 389 - 394**

Reviewer #2 (Remarks to the Author):

In the study entitled “Molecular epidemiology of *Ascaris lumbricoides* following multiple rounds of community-wide treatment: who infects whom?”, the authors sequenced 54 whole-genomes from individuals in a longitudinal cohort epidemiological study of transmission and drug treatment extending over 6 years. This study provided valuable insights for shaping future control policies in low prevalence settings, highlighting the importance of more targeted treatment of infection hotspots.

However, there are some points that the authors need to address to make the manuscript clearer:

1. Detailed explanation of “who infects whom”. – **amended – see lines 79 - 85**
2. The citation format is not standardized. – **amended to Nature publication portfolio guidelines**
3. References 26 and 30 are identical. – **amended**

Reviewer #3 (Remarks to the Author):

Nature Communications Review Jan 2025

This study describes the generation of whole genome sequence data from adult *Ascaris* worms obtained from people living in one community in Ethiopia enrolled in a longitudinal study investigating interventions to interrupt the transmission of soil-transmitted helminths and schistosome infections. By the time of worm collection, participants had received multiple rounds of mass drug administration over a 6-year period. The study links genomic and spatial data at a community level. It also explores regions of positive selection across the *A. lumbricoides* genome.

This is a very interesting study generating important new data and the analyses are carefully conducted. However, there are a couple of limitations which should be highlighted. Firstly, the worms which were analysed were obtained after standard MDA treatment with albendazole. This means that the sampled worms were sensitive to albendazole. There may well be other worms with reduced response to albendazole

circulating in the population which would not be sampled using this approach. This limitation should be mentioned in the Discussion and the results interpreted accordingly. Secondly, no samples were collected at the beginning of the study for comparison with samples collected after 6 years of MDA. This should be mentioned as a limitation in the Discussion. - **amended and addressed from lines 441 to 456.**

Other recommended revisions are as follows:

Abstract

Line 24 – insert “Ascaris worms obtained from” after “from” – **amended – See line 33**

Line 30 – should “no high” be “low”? – **amended – see line 38**

Introduction

Lines 35-39 – There are more recent estimates of the DALYs associated with intestinal nematode infections and number of people infected with Ascaris from the Global Burden of Diseases study, see the Institute for Health Metrics and Evaluation. Please update the figures accordingly. – **amended – See line 45 for the updated DALY’s references and figures**

Line 41 – replace “of” with “using” – **amended – see line 49**

Line 46 – perhaps “pre-SAC” is better than “pre-” – **amended – see line 54**

Line 50 – clarify “To achieve transmission, break a 2% prevalence must be achieved” – **amended – see line 58-59**

Line 59 – perhaps “is seen” or “exists” is better than “perturbs” – **amended – see line 66-67**

Line 69 – replace “renainging” with “remaining” – **amended – see line 77**

Line 70 – replace “result due to many by” with “may be attributed to” – **amended – see line 78**

Line 101- states “largest WGS study”, but Easton et al. 2020 paper performed more (n=68, I believe, albeit maybe at a lower coverage) - **amended**

Materials and methods

There is no description of the statistical methods used to analyse the epidemiological data. These should be included. – **amended - see lines 164 to 168**

It is not clear where the genomic data generated during this study will be made available to the scientific community. Details should be provided. – **All reads are available in SRA format on ncbi, accession numbers are made available in “supplementary table 2” this has been referenced in the text**

Most of the description of the bioinformatics methods is in the supplementary material. It would be useful to include more in the main body of the text, and signpost the supplementary material where needed, - **amended – the materials and methods have now been updated to include all analysis used in the bioinformatics workflow. The Supplementary text has been removed and reworded to be included within the**

main body of text.

Will any of the scripts used be available on Git-hub or an alternative repository? Line 106-107 – why were 20 households selected for worm collection? Why was Korke Doge kebele chosen for this study? – **amended – these have been added to a github and the address included in metadata of the manuscript – the public repository is found at: https://github.com/TobyLand/Ascaris_PopGen**

Line 109-110 – mention that 400 mg of albendazole was delivered as part of the community-wide MDA administered during the Geshiyaro project. Mention how many individuals had their stool examined for 5 days to collect worms and how these individuals were selected. Also state how worms were processed after collection. Were the worms washed? How? How were the worms stored and transported?

Line 110 – “Epidemiological data collection” should be on a new line – **amended – see line 124 to 137**

Line 116 – should it be seven rounds rather than four rounds of cMDA? A diagram summarising when the cMDA took place over the six-year period would be helpful. - **amended – see figure 1**

Line 129 – Where did DNA extraction take place? How long was the gap between worm collection and DNA extraction? Were the worms pre-washed/rinsed in any buffer or retained in any storage media? How large were the snip sections? What tissue was used for DNA extraction and this just somatic tissue or also germline? Was the cuticle removed before DNA extraction? How was decided which adult worms to sequence? It was mentioned in the supplementary material that DNA was extracted from 66 worms but 54 are mentioned in the main text. Please correct this discrepancy. Were any of the collected worms sexed, particularly the worms that were sent for sequencing? –

amended - see line 180 – 201 additional data has been added regarding worm collection, DNA extraction location and storage between lines 124 and 139

Line 130 – Was Qiagen MagAttract kit used according to manufacturer’s instructions? If so this should be mentioned. Otherwise, modifications should be described. – **amended – see line 189**

Line 132 – Apart from DNA quantification, was there any quality control to look at level of DNA fragmentation e.g. by TapeStation? – **amended see line 188 - 191**

Line 140 – “reads were aligned using BWA mem” could clarify here what they were aligned against (include accession numbers) – **amended see line 204 - 206**

Line 145 - The use of GATK VariantFiltration for filtering SNPs and indels is mentioned, but there is no information on specific filtering thresholds (e.g., minimum QUAL score or min depth). - **amended – see lines 210 - 215**

Line 152 – Explain why variants that were in strong linkage disequilibrium were removed before conducting population genomic analysis. **Amended – see lines 225 - 230**

Line 152 – Add abbreviation for principal component analysis (PCA) - **amended**

Line 153 – PLINK – missing the version number – **amended**

Line 154 – “1 to 20,” to “1 to 20.” – **amended**

Line 154 – should it be ADMIXTURE? If so, please include the version where possible -

amended

Line 157 – PIXY – version number missing - **amended**

Line 164 – change “principal component analysis” to PCA - **amended**

Line 166 - change “principal component analysis” to PCA - **amended**

Results

Line 170-175 – It could be clearer that prevalence was determined using faecal egg counts (mention the method used) vs identifying worms in faeces. – **amended – see lines 248 - 249**

Line 172 – suggest using “collection” rather than “expulsion” as worms were collected after MDA rather than after using a treatment to stimulate expulsion. - **amended**

Line 194 – the reference *A. lumbricoides* genome was used for alignment. Was any consideration given to using the reference *A. suum* genome which is higher quality and closely-related? – **amended – this can be seen in material and methods section, line 205-206**

Line 198 – Is the coverage range correct? The lower figure does not appear to match that in the supplementary data. - **amended**

Line 198 – Did you check to see if low read depth was consistent across the samples/ focused on specific loci? – **amended – there was no specific association with specific regions or loci exhibiting low coverage and has been mentioned in the text line 277-278**

Line 202 – “sample” should be “samples” - **amended**

Line 203 – use PCA abbreviation only - **amended**

Line 206 – remove question mark (sentence could be improved for clarity) – **amended**

Line 210 – should “establish” be “established”? Line 211 – insert space after

“(Supplementary Fig 4).” - **amended**

Line 212 – how was the subset of autosomal variants created? – **amended, added to material and methods section**

Line 217 - Should “population” be “populations”? - **amended**

Line 228 – “Adolescents” should be “adolescents” - **amended**

Line 229 – “Adults” should be “adults” - **amended**

Line 245 - “highlighted reduced geneflow between worms from compared from within households compared to between” - revise for clarity - **amended**

Line 278 – Remove “Within these regions we identified a single locus containing beta tubulin gene.” This repeats earlier text. - **amended**

Line 280-282 – There is quite a bit of discussion in the literature as to whether beta-tubulins are linked to reduced efficacy in *A. lumbricoides*. The link is clear in clade 5 nematodes infecting ruminants and horses but there is no clear evidence in ascarids as of yet. This warrants further discussion but this is probably better placed in the Discussion section rather than the Results. – **amended – this has been included as a point in the discussion – see lines 434 - 445**

Discussion

Line 311 – change “genomic” to “Genomic” - **amended**

Line 316 – Please clarify how the data indicates a shrinking population of *A. lumbricoides* across Korke Doge. - **amended**

Line 322 – What about environmental reservoirs? – **amended within a rewritten section across lines 403 - 424**

Line 324-325 – Sentence could be made clearer - **amended**

Line 331 – change “If variants associated” to “If variants are associated” - **amended**

Line 349 – I think somewhere it needs to be mentioned that in other communities using faeces as fertiliser is a major contributor human transmission. Therefore, more molecular studies of this nature will be required to actively support control initiatives. You could mention pig-human transmission in this regard, also. – **amended – this has been included as a discussion point regarding reasoning behind transmission foci – see lines 393 - 402**

Figure 1 – Could include a scale - **amended**

Figure 2. - Include total numbers of people tested.

Figure 2 – “a” is bold but “b” is not. - **amended**

Figure 2 - “The target 2% prevalence of *A. lumbricoides* is shown as a grey dashed line”.

This is not very visible – **As this is a Geshiyaro-only project goal and not a WHO prescribed threshold of prevalence to break transmission it has been removed as to not cause confusion**

Figure 2 – You could include the WHO guidelines of ≤ 4999 epg – **amended in figure footnote**

Table 1. Include total numbers for each time period. – **amended table**

Figure 3 – Reword “between the 54 *A. lumbricoides* produced sampled for WGS” – **amended**

Figure 6 – higher resolution image required – **amended**

Supplementary text

Should supplementary text be referenced in the main text? – **Following consultation with editor the supplementary text has now been included in the main text as removed as supplementary**

DNA extraction and sequencing – Change “snips” to snip. Was the Invitrogen Qubit high sensitivity or broad-spectrum assay used? Ensure *A. lumbricoides* is italicized correctly throughout. Was 100bp or 150bp used for Illumina sequencing?

I understand that this is the supplementary text, but include version numbers of software where possible. – **Version numbers have now been included in all mention of software toolkits**

“five years of treatment” mentioned at the top of page 2. Is it five or six years of treatment? - **amended – clarification has been added in the text, the worm expulsion took place in year five of the project following four rounds of MDA. This**

has been clarified with a gantt chart inclusion of the full control program timeline.

Text has been amended at lines 252 - 255

The supplementary text mentions that DNA was extracted from 66 worms and that 68 DNA samples were sequenced. This contrasts with 54 worms in the main text.

In "Spatial population genomics" paragraph, the first sentence is not a proper sentence.

There is an extra full stop at the end of the paragraph. – **amended**

Formatting of Supplementary Figure 4 could be improved to improve readability. –

amended for greater resolution

Supplementary Figure 5 – Check this sentence: "All three plots taken in the round indicate that three genetic clusters exist in three genetic clusters". Also "e plot

represented" should be "e plot represents". – **amended**

Response to reviewers

I would like to thank reviewer 3 again in their response to the manuscript we have addressed these issues and have been outline below:

A couple of minor issues

There are a few sentences that are confusing i.e line 219, so please read carefully.

This has been amended (now lines 478 to 485)

Check the figure numbers as the legend jumps from 5 to 7a.

Amended